# *Lactobacillus acidophilus* C4 ameliorates constipation in mice

Han Yan,[1] Qianying Jia,[1] Jing Liu,[1] Xinping Hua,[1] Yihong Cui,[1] Wenwen Jian,[1] Yu Tan,[1] Ochieng Samuel Oduor,[1] Privillia Thomas,[1] Mwansa Kawimbe,[1] Yuan Wu,[2] He Lu,[1] Tianle Gu,[1] Zeng Tu[1]

**ABSTRACT** Functional constipation (FC) is a common gastrointestinal disorder that can seriously affect patients' quality of life. Probiotics have emerged as a promising strategy for preventing and managing gastrointestinal diseases. Previously, we isolated *Lactobacillus acidophilus* C4 from the feces of a healthy child, and here, we investigated its probiotic efficacy against FC in a mouse model. Prophylactic treatment with *L. acidophilus* C4 prior to FC onset significantly reduced impairment of defecation function and increased intestinal transit rate. *L. acidophilus* C4 also increased the expression of tight junction proteins ZO-1 and occludin, suggesting a protective effect on the intestinal barrier. In addition, *L. acidophilus* C4 was able to modulate the inflammatory response by decreasing the pro-inflammatory factor IL-1β while increasing the anti-inflammatory cytokine IL-10. These results reveal that *L. acidophilus* C4 can improve constipation through anti-inflammation and restoration of intestinal barrier function, which can be a candidate strain for the development of novel interventions for constipation.

**IMPORTANCE** Functional constipation (FC) requires novel therapeutic approaches. Probiotics have attracted attention in the prevention and management of gastrointestinal diseases; however, their efficacy and mechanisms of action against FC need to be verified. This study provides experimental evidence for this purpose, supporting its potential as a non-invasive and well-tolerated intervention. The core of this study is *Lactobacillus acidophilus* C4, which exhibits high compatibility with the human intestinal microenvironment and lays a foundation for clinical research. *L. acidophilus* C4 can enhance intestinal peristalsis, protect the intestinal barrier (by upregulating tight junction proteins ZO-1 and occludin), and regulate inflammation (by balancing the pro-inflammatory cytokine *IL-1β* and anti-inflammatory cytokine *IL-10*). This strain provides a new direction for research on probiotic-based treatment of FC.

**KEYWORDS** probiotics, *Lactobacillus acidophilus*, functional constipation, intestinal transit rate

F unctional constipation (FC) is a common gastrointestinal disorder that has become a global public health problem (1). FC is characterized by decreased frequency of bowel movements, dry stools, painful or difficult defecation, and prolonged gastrointestinal transit time (2, 3). The prevalence of constipation ranges from 2% to 30% and is particularly high in children and the elderly, with a higher prevalence in women than in men (4, 5). Prolonged retention of feces in the colon leads to over-absorption of water, formation of hardened feces, and slowing down of intestinal motility, resulting in an imbalance of intestinal microbiota and abdominal pain. As a prevalent gastrointestinal disorder, functional constipation can seriously affect patients' quality of life. Conventional clinical treatment relies mainly on laxatives and prokinetic agents, but long-term use can induce drug dependence (6). These limitations highlight the urgent need to develop safer and sustainable treatment options.

Address correspondence to Tianle Gu, gutianle@cqmu.edu.cn, or Zeng Tu, tuzeng@cqmu.edu.cn.

Han Yan and Qianying Jia contributed equally to this article. Author order was determined in order of increasing seniority.

The authors declare no conflict of interest.

Nutritional therapy for constipation involves the intake of dietary fiber-rich foods (e.g., fruits, vegetables, whole grains), which can effectively shorten the intestinal transit time by increasing the volume of feces and accelerating the transit of small intestinal content (7–10). Recent studies have further revealed that probiotics represented by *Lactobacillus* spp. play a key role in regulating intestinal microecology and exert a positive effect on alleviating constipation (11). Their mechanisms of action include enhancing intestinal motility, improving fecal consistency to increase bowel frequency, and regulating the structure of intestinal microbiota. Additionally, they improve intestinal mucosal barrier function and modulate host immune response to achieve multi-targeted interventions; and hence, generating biologically active metabolites, such as short-chain fatty acids, which indirectly modulate intestinal motility rhythms through the enteric-encephalic axis (12–14).

*Lactobacillus acidophilus* was first isolated from infant feces at the beginning of the 20th century and has become one of the probiotic strains in the field of functional foods (15). We successfully isolated a new strain of *L. acidophilus,* designated as C4, which exhibits potent *in vitro* probiotic properties, including acid and bile tolerance and intestinal adhesion capability (16). Furthermore, we confirmed that this strain can significantly alleviate the symptoms of ulcerative colitis through inhibition of inflammatory factors (IL-6, TNF-α) and protect intestinal barrier function (16). Thus, we speculate that this strain may have a therapeutic function in constipation. In this study, we investigated the effects of *L. acidophilus* C4 on FC in a mouse model and characterized the molecular mechanism of regulating intestinal motility.

## MATERIALS AND METHODS

### Materials and diet

The *L. acidophilus* C4 used in this study was isolated from a previous work done by our research group, whereby we demonstrated its protective effects against dextran sulfate sodium-induced murine colitis. As described in our earlier report, this strain was initially obtained through selective screening of fecal specimens collected from healthy children (16). Loperamide capsules were purchased from Xi'an Janssen Pharmaceutical (9). Activated charcoal and gum Arabic were purchased from Tianli Chemical Reagent. Four percent paraformaldehyde was purchased from Biosharp Biotechnology.

### Preparation of probiotics

*L. acidophilus* C4 and *Lactobacillus rhamnosus* LGG stored at −80°C were thawed on ice, a loop of the bacterial solution was inoculated onto the De Man, Rogosa and Sharpe Agar (MRS Agar) using the three-zone streaking method and incubated in an anaerobic chamber at 37°C for 24–48 h. A single colony was picked from the culture plate and transferred to the MRS liquid medium and incubated in anaerobic conditions for 12 h. A 250 µL aliquot of this culture was inoculated into 50 mL of fresh MRS liquid medium and anaerobically incubated for 12 h. After adjusting the bacterial concentration to $1 \times 10^9$ CFU/mL, the culture was centrifuged at 4°C at 4,000 rpm for 5 min. The bacterial pellet was washed three times with PBS and resuspended in saline (9, 17).

### Mouse model and experimental design

Healthy female KM mice (6 weeks old) were purchased from Hunan Slake Jinda Laboratory Animal Co., Ltd. (license no.: SCXK [Xiang] 2021-0002). All experimental animals were maintained in a controlled environment at 22°C–26°C with a 12 h light/12 h dark circadian cycle, and mice had *ad libitum* access to standard laboratory food and sterile water throughout the experiment. Eight mice were housed per cage, fed a standard diet (Chongqing Enbi Biotechnology, China), with daily changes of feed, water, and sterilized bedding during the experiment.

To evaluate the effect of *L. acidophilus* C4 on constipation in mice, 16 mice were randomly assigned to two groups: normal control (Ctrl) and C4 control (C4) after a 7 day acclimatization period. Over 28 consecutive days, the C4 group received daily gavage of 200 µL probiotic suspension ($1 \times 10^9$ CFU/mL) at 9:00 am, while the Ctrl group received a gavage of equal volume of saline (Fig. 1A). Additionally, from day 15 onward, both groups were gavage-fed with an equal volume of normal saline (NS) at 3:00 pm daily. During the experiment, the activity, mental state, and the mice fur color in each group were monitored daily. Food and water intake, as well as individual body weights, were recorded daily at a fixed time to ensure data consistency (7).

To induce constipation and evaluate probiotic efficacy, 32 mice were randomly allocated to four groups after a 7 day acclimatization period: Ctrl, constipation model (LOP), C4 intervention (LOP+C4), and LGG positive control (LOP+LGG). *L. rhamnosus* LGG (standard strain, ATCC 53103) was used as a well-established positive control in this study (18, 19). The constipation-alleviating efficacy of *L. rhamnosus* LGG has been verified by numerous clinical studies, and its mechanisms of action (including regulating intestinal microbiota, promoting intestinal motility, etc.) have been well-defined, making it a classic reference strain in the field of probiotic intervention for constipation (18, 19). Therefore, the inclusion of *L. rhamnosus* LGG enabled a direct comparative assessment of the potency of the novel *L. acidophilus* C4. Over 28 consecutive days, the LOP+C4 and LOP+LGG groups received a daily gavage of 200 µL of their respective probiotic suspensions ($1 \times 10^9$ CFU/mL) at 9:00 am, while the Ctrl and LOP groups received equal volumes of NS. From day 15 onward, all groups except Ctrl received a gavage of loperamide (10 mg/[kg·bw]) at 3:00 pm daily for 14 days to induce constipation (17) (Fig. 1B). At the end of the experiment, mice were euthanized. The anesthetic compound Tribromoethanol (250 mg/kg, 300 µL) was administered intraperitoneally to the mice. After complete anesthesia, cervical dislocation was performed to euthanize the

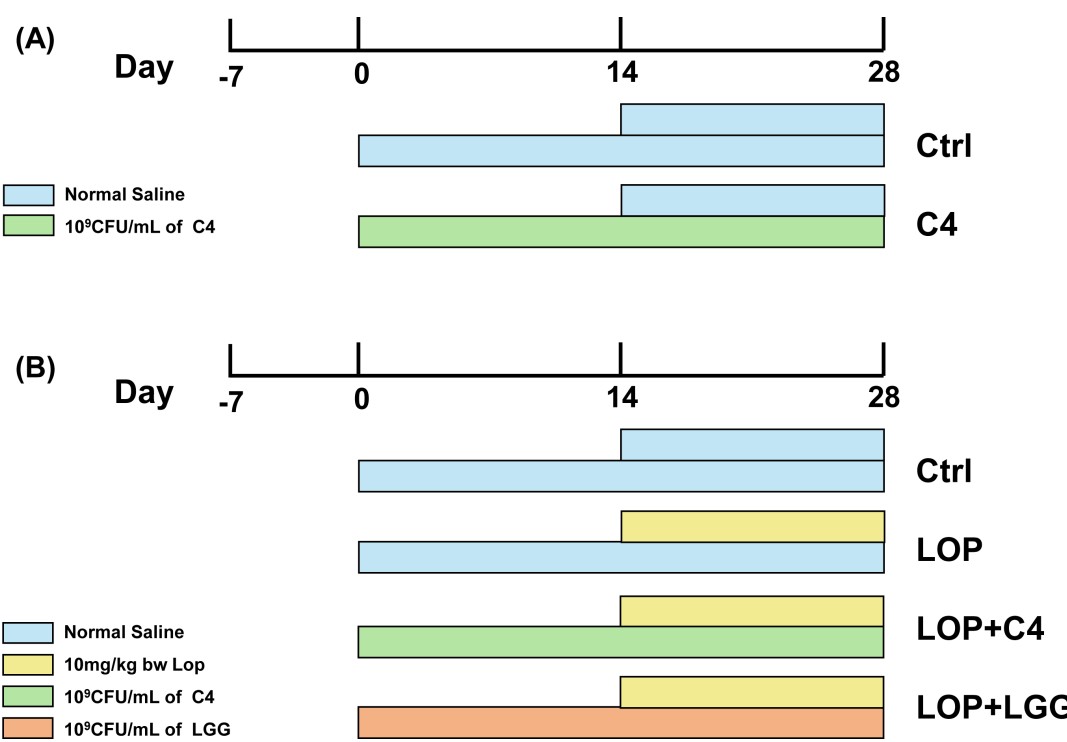

**FIG 1** Experimental design. (A) Sixteen female mice were randomly assigned to two groups: normal control (Ctrl) and C4 control (C4); the C4 group received daily gavage of *L. acidophilus* C4 suspension, while the Ctrl group was gavaged with normal saline. (B) Thirty-two female mice were randomly divided into four groups: normal control (Ctrl); constipation model (LOP), C4 intervention (LOP+C4), and positive control (LOP+LGG). The LOP+C4 and LOP+LGG groups received daily gavage of *L. acidophilus* C4 or *L. rhamnosus* LGG, respectively, while the Ctrl and LOP groups received normal saline. From day 15 onward, all groups except the Ctrl group were gavaged with loperamide at a daily dose of (10 mg/[kg·bw]) to induce constipation.

mice. Gastrointestinal tissues and intestinal contents were collected for further analysis. All samples were stored at −80℃ for subsequent experiments. Animal carcasses were placed in yellow biohazard bags and stored in a dedicated 4℃ freezer for experimental animal remains (17).

## Mouse defecation test

During the experiment, on day 7, day 14, day 21, and day 27, respectively, each mouse was transferred to a clean, empty cage and housed for 6 h. During this period, fecal samples were collected, counted, and weighed for subsequent analysis of fecal indicators (17).

## Whole gut transit time

To prepare the ink indicator, 100 g of gum Arabic and 800 mL of water were boiled until the solution became clear. Then, 50 g of activated charcoal powder was added, and the mixture was boiled three times. After cooling, the solution was diluted to 1,000 mL with water, stored at 4℃, and thoroughly mixed before use.

On day 6, day 13, day 20, and day 26, mice were starved for 16 h with free access to water. Then, mice in the model groups were administered loperamide (10 mg/[kg·bw]) via gavage, while the normal control group received saline. Thirty minutes later, all mice received a gavage of 200 μL of the ink indicator, and the time from ink administration to the appearance of dark-colored feces was recorded (9, 17).

## Measurement of intestinal propulsion rate

On day 27, after overnight fasting (16 h) with water available, all mice received 200 μL of the ink indicator by gavage and were returned to cages with free access to food and water. Thirty minutes after gavage, mice were anesthetized with light ether. The entire small intestine (from pylorus to cecum) was harvested. The distance traveled by the ink and the total length of the small intestine were measured, and the small intestinal propulsion rate was calculated as (Distance traveled by ink / Total length of small intestine) × 100% (7, 9, 17).

## Real-time fluorescence quantitative PCR (RT-qPCR)

Total RNA from colon tissue was extracted with TRIzol (15596026CN, Invitrogen). cDNA was generated using a reverse transcription kit (RR047A, TaKaRa). cDNA products were subjected to RT-qPCR in an ABI 7500 instrument (Applied Biosystems) with gene-specific primers (Table 1). The amplification reaction conditions were: 95℃ 3 min, 95℃ 5 s, 60℃ 30 s × 40 cycles. The β-*actin* gene was used as an internal reference gene for normalization, and the expression level of the target gene was calculated by the $2^{-\Delta\Delta ct}$ method.

## Western blotting

Following euthanasia, mice were dissected in 4℃ PBS. Portions of colon tissue were harvested and immediately frozen in liquid nitrogen before being stored at −80℃. Colon tissue samples were homogenized with added lysis buffer (P0013C, Beyotime), and the homogenates were centrifuged at 12,000 rpm at 4℃ for 20 min. The resulting supernatant was collected as the cytoplasmic protein extract.

Protein concentration was determined using the BCA Protein Assay Kit (P0012S, Beyotime). Twenty micrograms of protein was mixed with SDS-PAGE loading buffer, denatured at 100℃ for 10 min, and separated by 10% SDS-PAGE (PG112, Yamei Biology). Proteins were transferred to PVDF membranes (E801-1, Vazyme). Membranes were blocked in TBST containing 5% skimmed milk powder for 1.5 h at room tempera-ture, then incubated overnight at 4℃ with primary antibodies against ZO-1 (TA5145F, 1:1,000; Abmart) and occludin (AWA10233, 1:1,000; Abiowell). After three washes with

**TABLE 1** RT-qPCR primers

| Primer name | Sequences |
| --- | --- |
| IL-1β-F | 5′-GAAATGCCACCTTTTGACAGTG-3′ |
| IL-1β-R | 5′-TGGATGCTCTCATCAGGACAG-3′ |
| AQP3-F | 5′-TGCCTTGCGCTAGCTACTTT-3′ |
| AQP3-R | 5′-GCCACAGCCAAACATCACAA-3′ |
| ZO-1-F | 5′-CTTCTCTTGCTGGCCCTAAAC-3′ |
| ZO-1-R | 5′-TGGCTTCACTTGAGGTTTCTG-3′ |
| Occludin-F | 5′-CACACTTGCTTGGGACAGAG-3′ |
| Occludin-R | 5′-TAGCCATAGCCTCCATAGCC-3′ |
| β-actin-F | 5′-CACTGTCGAGTCGCGTCCA-3′ |
| β-actin-R | 5′-CATCCATGGCGAACTGGTGG-3′ |

TBST, membranes were incubated with HRP-conjugated Goat Anti-Rabbit IgG (H+L) secondary antibody (SA00001-2, 1:10,000; Proteintech) or directly with HRP-conjugated recombinant anti-β-actin antibody (ZB15001-HRP, 1:1,000; Servicebio) for 1 h at room temperature. Following three additional washes, target bands were visualized using the Ultra-Sensitive ECL Chemiluminescence Detection Kit (SW134-02; SEVEN) with a Bio-Rad ChemiDoc imaging system.

## Statistical analysis

Data analyses were performed using GraphPad Prism version 10. Data are expressed as mean ± standard deviation (SD). Differences between groups were evaluated using one-way ANOVA or two-way ANOVA, as appropriate. $P < 0.05$ was considered statistically significant.

## RESULTS

### Acidophilus C4 ameliorates loperamide-induced constipation in mice

To assess the effect of *L. acidophilus* C4 on constipation in mice, we first evaluated its impact on the mice's food intake, water intake, and body weight. The results showed that both the Ctrl and C4 (administered with *L. acidophilus* C4) groups experienced a steady increase in body weight throughout the experimental period (from 28.37 ± 1.43 g to 37.45 ± 2.27 g) (Fig. 2A). Additionally, there were no significant differences between the Ctrl and C4 groups in average daily food intake (6.45 ± 0.12 g vs 5.34 ± 0.08 g) (Fig. 2B) or water intake (6.95 ± 0.39 g vs 7.06 ± 0.03 g) (Fig. 2C) ($P > 0.05$). These results indicated that *L. acidophilus* C4, at the dosage used in this experiment, had no significant impact on the physiological status indicators (body weight, food intake, and water intake) of the mice.

To investigate the effect of probiotics on functional constipation, mice were first treated with different probiotic strains (*L. rhamnosus* LGG was included as a positive control, as previous studies have shown that it can ameliorate constipation) (20), followed by induction of functional constipation using loperamide—an FDA-approved antidiarrheal agent widely used to establish animal models of functional constipation. The administration of probiotics and loperamide had no significant effect on body weight gain (37.16 ± 1.89 vs 37.71 ± 1.86 vs 36.76 ± 1.03 vs 38.11 ± 1.76), food, or water intake (Fig. 2D through F).

Functional constipation is characterized by key symptoms such as reduced defecation frequency, hardened stools, and prolonged defecation time (21). To investigate the effect of probiotics on defecation function in constipated mice, we assessed several parameters: whole gut transit time, fecal output, fecal water content, and fecal wet weight. In healthy mice, *L. acidophilus* C4 had no significant effect on whole gut transit time ($P > 0.05$) (Fig. 3A), fecal output within 6 h (Fig. 3B), fecal water content (Fig. 3C), or fecal wet weight (Fig. 3D) (all $P > 0.05$).

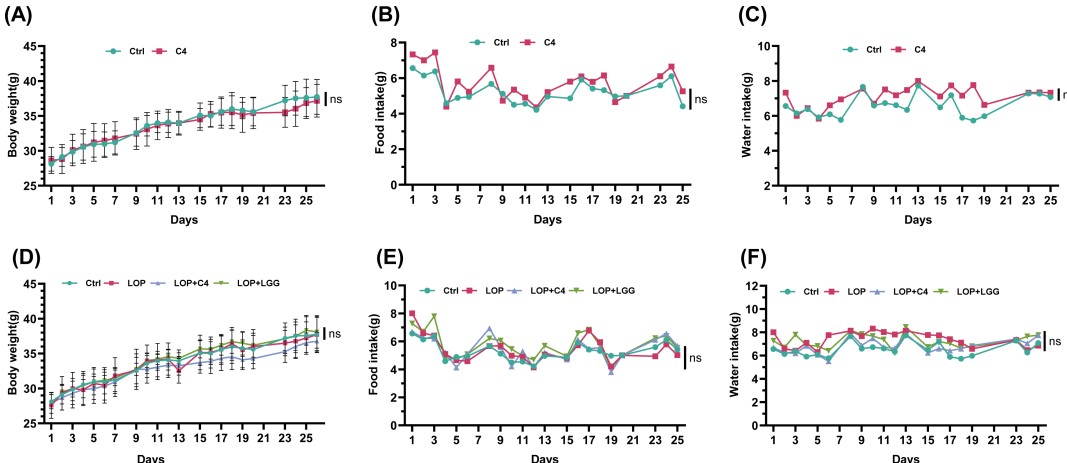

**FIG 2** *L. acidophilus* C4 had no effect on the physiological status indicators of mice. (A–C) Mice were gavaged with *L. acidophilus* C4 or normal saline. The (A) body weight, (B) food intake, and (C) water intake were measured at the indicated time points. (D–F) Mice were gavaged with *L. acidophilus* C4, *L. rhamnosus* LGG, or normal saline. On day 15, mice were treated with loperamide or not. The (D) body weight, (E) food intake, and (F) water intake were recorded at the indicated time points. Ctrl, normal control group. C4, C4 intervention normal mice. LOP, loperamide-induced model mice. LOP+C4, loperamide-induced model mice prevented with *L. acidophilus* C4. LOP+LGG, loperamide-induced model mice prevented with *L. rhamnosus* LGG. Data are expressed as mean ± SD (*n* = 8). ns, not significant.

In mice with loperamide-induced constipation (constipation was induced via loperamide administration from days 15 to 28 of the experiment), obvious differences between some groups were observed on day 21. By day 27, the time to first discharge of black stool was significantly longer in the LOP group than in the Ctrl group ($P <$ 0.05) (Fig. 3E). Conversely, the number of fecal pellets excreted within 6 h ($P < 0.05$) (Fig. 3F), fecal water content (Fig. 3G), and fecal wet weight (Fig. 3H) in the LOP group were significantly lower than those in the Ctrl group, confirming successful induction of functional constipation in mice. Pretreatment with *L. rhamnosus* LGG prior to loperamide administration significantly reversed these phenotypic changes, consistent with previous findings (20). Similarly, pretreatment with *L. acidophilus* C4 significantly shortened the whole gut transit time (Fig. 3E), increased fecal output within 6 h (Fig. 3F), elevated fecal water content (Fig. 3G), and increased fecal wet weight (Fig. 3H). These results indicate that pretreatment with *L. acidophilus* C4 can significantly ameliorate loperamide-induced constipation in mice.

## *L. acidophilus* C4 enhances intestinal motility

In general, intestinal peristalsis slows down in patients with constipation. Intestinal motility is mainly responsible for the movement of intestinal contents (9). The ink propulsion rate can reflect intestinal transit function. To investigate the effect of *L. acidophilus* C4 on intestinal motility in constipated mice, we assessed its impact using the intestinal ink propulsion rate assay. The total length of the small intestine showed no significant difference among groups (Fig. 4A and B). The intestinal ink propulsion rate in the LOP group was significantly lower than that in the Ctrl group (47.22 ± 0.39 vs 74.67 ± 3.18) (Fig. 4C), confirming the successful induction of constipation.

Pretreatment with *L. acidophilus* C4 prior to loperamide administration increased the intestinal ink propulsion rate compared to the LOP group (72.16 ± 2.83 vs 47.22 ± 0.39) (Fig. 4A and C). However, *L. rhamnosus* LGG on rate in constipated mice pretreatment had no significant effect on the intestinal ink propulsion rate in constipated mice (48.07 ± 7.78 vs 47.22 ± 0.39) (Fig. 4A and C). These results suggest that *L. acidophilus* C4 may ameliorate loperamide-induced constipation by enhancing intestinal motility.

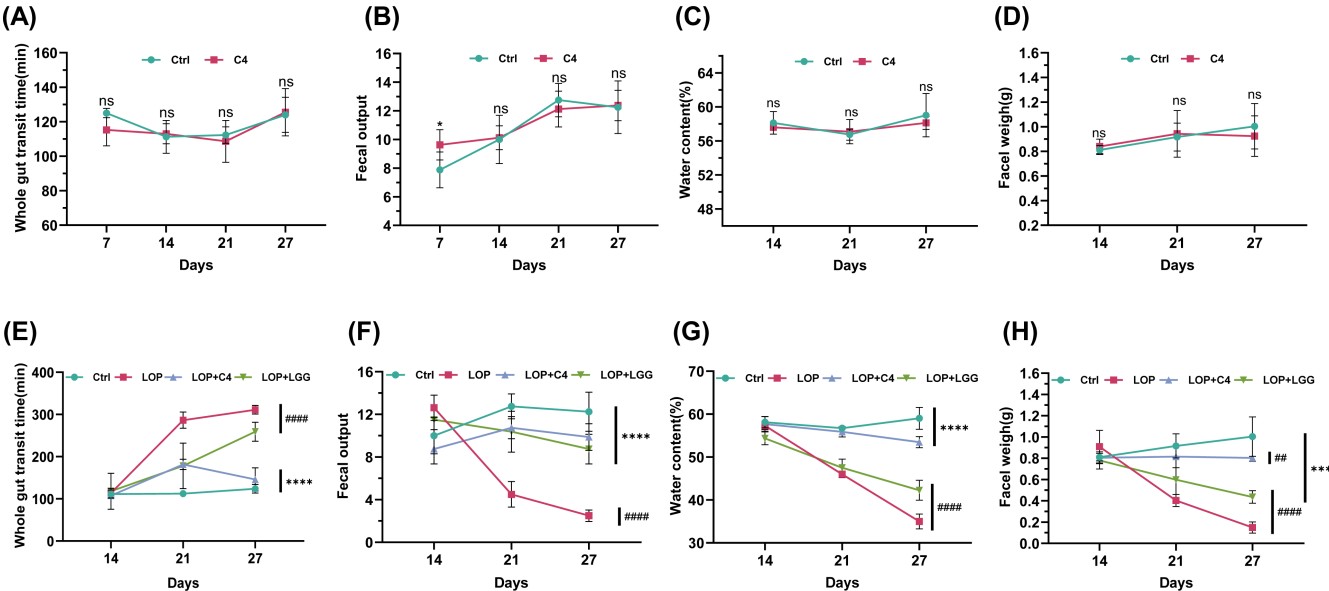

FIG 3 *L. acidophilus* C4 ameliorates loperamide-induced constipation in mice. (A–D) Mice were treated as in Fig. 1A. On day 7, day 14, day 21, and day 27, mice in each group were fasted, and all feces excreted within 6 h were collected. The (A) whole gut transit time, (B) fecal output, (C) fecal water content, and (D) fecal weight of healthy mice were measured. (E–H) Mice were treated as in Fig. 1B. On day 7, day 14, day 21, and day 27, mice in each group were fasted, and all feces excreted within 6 h were collected. The (E) whole gut transit time, (F) fecal output, (G) fecal water content, and (H) fecal weight were measured. Ctrl, normal control group. C4, C4 intervention normal mice. LOP, loperamide-induced model mice. LOP+C4, loperamide-induced model mice prevented with *L. acidophilus* C4. LOP+LGG, loperamide-induced model mice prevented with *L. rhamnosus* LGG. Data are expressed as mean ± SD ($n$ = 8). Compared with the loperamide-induced model mouse group, (****) $P < 0.0001$. Compared with the normal control group, (##) $P < 0.01$, (####) $P < 0.0001$. ns, not significant.

## *L. acidophilus* C4 mitigates intestinal barrier dysfunction and restores colonic water homeostasis in constipated mice

Constipation is associated with intestinal barrier impairment and dysregulated intestinal water content (22). Our previous findings demonstrated that *L. acidophilus* C4 enhances intestinal motility in constipated mice; thus, we hypothesized that *L. acidophilus* C4 may exert a protective effect on the intestine during constipation. The intestinal mucosal barrier maintains the normal intestinal microecological environment, and ZO-1 and occludin are core structural proteins of colonic mucosal tight junctions, whose expression levels directly reflect mucosal barrier integrity (16, 23). To evaluate the effect of *L. acidophilus* C4 on intestinal integrity, we measured the expression of these

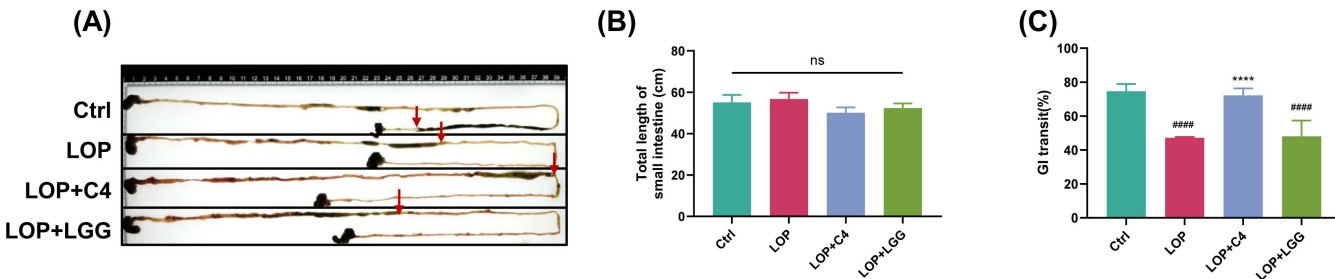

FIG 4 *L. acidophilus* C4 promotes small intestinal propulsion rate in constipated mice. Mice were treated as in Fig. 1B. On day 27 of the experiment, fasted mice were gavaged with an ink indicator. Thirty minutes later, the mice were dissected, and the entire small intestine from the pylorus to the cecum was harvested. Then, (A) photos of the small intestine with ink (to record the distribution of ink in the small intestine) were taken, (B) the total length of the small intestine was measured, and (C) the gastrointestinal transit rate was calculated. Ctrl, normal control group. LOP, loperamide-induced model mice. LOP+C4, loperamide-induced model mice prevented with *L. acidophilus* C4. LOP+LGG, loperamide-induced model mice prevented with *L. rhamnosus* LGG. Data are expressed as mean ± standard deviation (SD) ($n$ = 6). Compared with the loperamide-induced model mouse group (****) $P < 0.0001$, compared with the normal control group, (####) $P < 0.0001$, ns, not significant.

tight junction-related genes and proteins in mouse intestines. The results showed that, compared with the Ctrl group, the transcript levels of both *ZO-1* and *occludin* genes were significantly downregulated in the LOP group ($P < 0.05$). In contrast, *L. acidophilus* C4 intervention significantly rescued the expression of *ZO-1* and *occludin* genes ($P < 0.05$), indicating amelioration of constipation-induced intestinal barrier dysfunction (Fig. 5A and B). At the protein level, ZO-1 and occludin expression were also significantly downregulated in the LOP group ($P < 0.05$), while the LOP+C4 group exhibited significantly higher ZO-1 and occludin protein levels compared with the LOP group ($P < 0.05$), further confirming the restoration of barrier function (Fig. 5C). Collectively, these results suggest that *L. acidophilus* C4 may have a protective effect on the intestinal mucosal barrier structure.

Aquaporin (AQP) channels play a central role in regulating colonic fluid homeostasis (22). Overexpression of AQPs promotes constipation development by enhancing colonic water reabsorption, which leads to a significant reduction in fecal water content (22). AQPs are primarily expressed in human colonic epithelial cells (24). We therefore determined the mRNA expression of *AQP3* in mouse intestines. Compared with the Ctrl group, the mRNA level of *AQP3* was significantly upregulated in the LOP group ($P < 0.05$) (Fig. 5D). However, both *L. acidophilus* C4 and *L. rhamnosus* LGG interventions significantly downregulated *AQP3* expression ($P < 0.05$) (Fig. 5D). Notably, the inhibitory effect on *AQP3* expression was stronger in the LOP+C4 group than in the LOP+LGG group ($P < 0.05$) (Fig. 5D). These results indicate that *L. acidophilus* C4 can restore colonic water metabolism balance by specifically regulating *AQP3* expression.

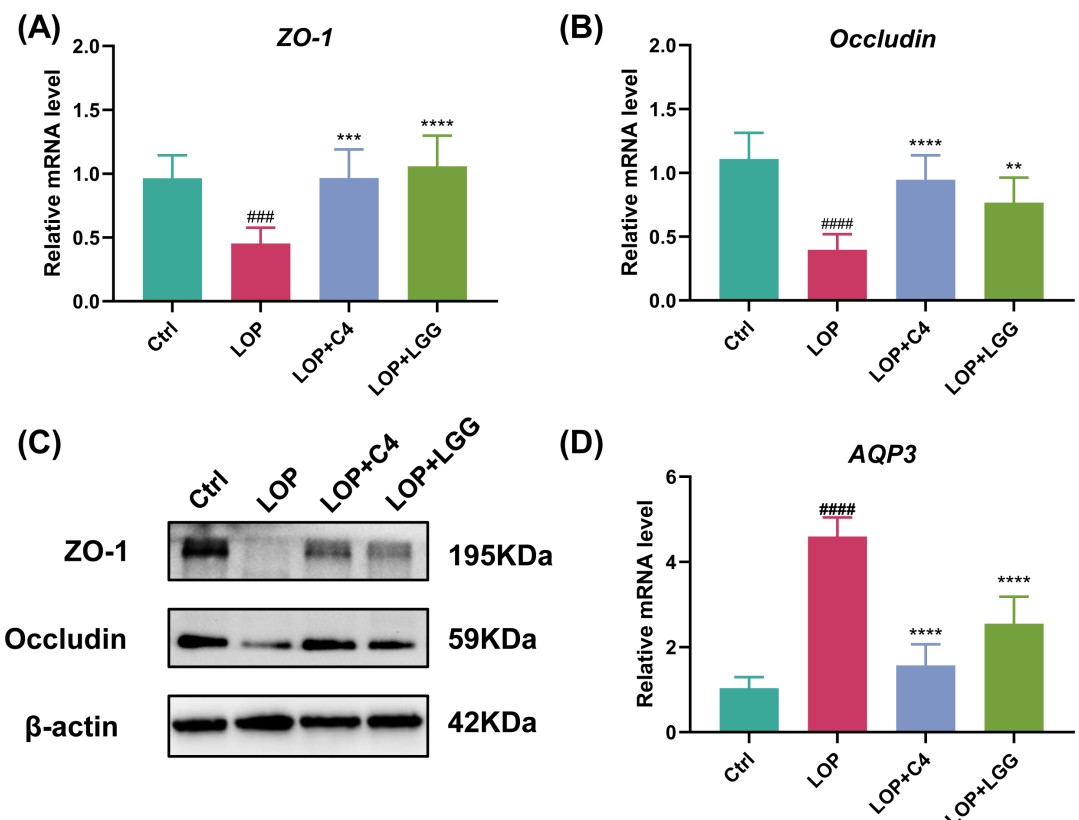

**FIG 5** *L. acidophilus* C4 mitigates intestinal barrier dysfunction and restores colonic water homeostasis in constipated mice. Mice were treated as in Fig. 1B. On day 27 of the experiment, mice were dissected, and colon tissues were harvested. The (A) *ZO-1*, (B) *occludin*, and (D) *AQP3* mRNA levels were detected by RT-qPCR, and the (C) ZO-1 and occludin protein expressions were detected by Western blotting. Data are expressed as mean ± SD ($n = 6$). Compared with the loperamide-induced model mouse group, (**) $P < 0.01$, (***) $P < 0.001$, (****) $P < 0.0001$, and compared with the normal control group, (####) $P < 0.0001$, is significant.

## *L. acidophilus* C4 attenuates constipation-associated intestinal inflammation

FC patients are often accompanied by intestinal inflammation, which in turn induces changes in inflammatory factor levels (23). The elevation degree of pro-inflammatory factors is positively correlated with the severity of intestinal inflammation (23). Interleukin-1β (IL-1β) is a key early pro-inflammatory factor in the body that activates acute-phase inflammatory responses, and its expression level can serve as an indicator for assessing the degree of inflammatory injury (23, 24). To evaluate the severity of intestinal inflammation in our model, we detected the expression mRNA level of *IL-1β* in the colon. As shown in Fig. 6A, the *IL-1β* mRNA level in colon tissue of the LOP group was significantly higher than that in the Ctrl group ($P < 0.05$), confirming that the constipation model successfully triggered an intestinal inflammatory response, which is consistent with previous reports (23). Notably, the *IL-1β* mRNA levels in both the LOP+C4 group and the LOP+LGG group were significantly lower than those in the LOP group ($P < 0.05$), indicating that *L. acidophilus* C4 and *L. rhamnosus* LGG effectively alleviated constipation-induced intestinal inflammatory injury.

Interleukin-10 (IL-10) is a critical anti-inflammatory factor that exerts immunomodulatory effects by inhibiting the secretion of pro-inflammatory factors (23). To further explore the effect of *L. acidophilus* C4 on intestinal inflammation during constipation, we measured the expression of *IL-10* in the colon. As shown in Fig. 6B, *IL-10* expression in the LOP group was significantly lower than that in the Ctrl group ($P < 0.05$), suggesting that the anti-inflammatory response was impaired in constipated mice. Importantly, compared with the LOP group, the *IL-10* mRNA level in the LOP+C4 group was significantly elevated ($P < 0.05$), indicating that *L. acidophilus* C4 can regulate intestinal immune homeostasis. These results collectively demonstrate that *L. acidophilus* C4 improves the imbalance in the *IL-1β/IL-10* expression ratio, which may contribute to the alleviation of constipation-associated intestinal inflammation by bidirectionally regulating the balance between pro-inflammatory and anti-inflammatory factors.

## DISCUSSION

Constipation is a common multifactorial syndrome with a global prevalence of 15.3%, with core symptoms of reduced fecal water content, decreased defecation rate (<3 times/week), and dyspareunia (Bristol classification ≤2) (25). Its etiology involves multidimensional pathophysiological changes such as abnormal intestinal neuromodulation, dyskinesia, and dysbiosis, which significantly reduces the quality of life of patients and induces anxiety and depression (26, 27). Epidemiological data show that about

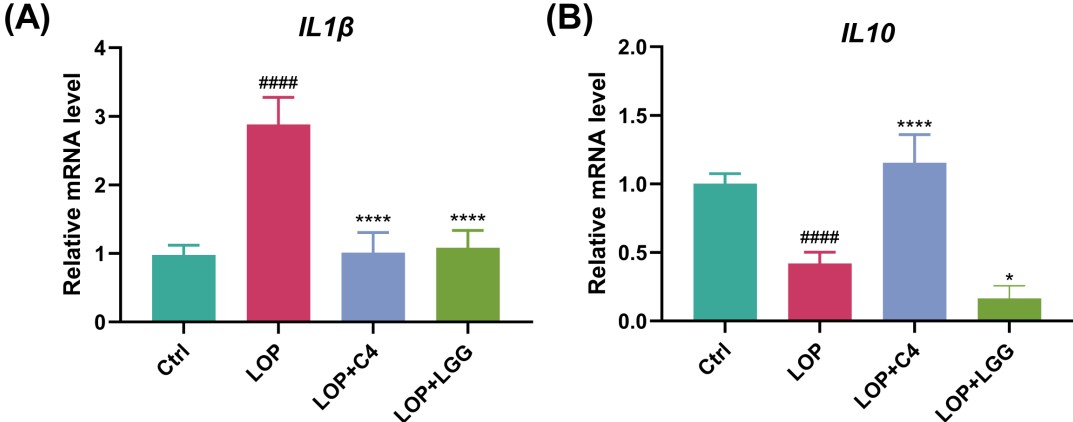

**FIG 6** *L. acidophilus* C4 attenuates constipation-associated intestinal inflammation. Mice were treated as in Fig. 1B. On day 27 of the experiment, mice were sacrificed and dissected, and colon tissues were harvested. The (A) *IL-1β* and (B) *IL-10* mRNA levels were measured by RT-qPCR. Data are expressed as mean ± SD ($n = 6$). Compared with the loperamide-induced model mouse group, (*) $P < 0.05$, (****) $P < 0.0001$, compared with the normal control group, (####) $P < 0.0001$, is significant.

60%–70% of constipation cases are FC (Rome IV criteria) (16). Recent studies have shown that an imbalance in the homeostasis of intestinal microbiota is a key causative factor of functional constipation (28, 29). Compared with the normal intestinal microbiota, the change of intestinal microbiota in patients with functional constipation is mainly manifested as a decrease in the abundance of beneficial bacteria such as *Bifidobacterium* and *L. acidophilus*, and an increase in the number of conditionally pathogenic bacteria such as Enterobacteriaceae (20, 29, 30). This disturbance of the intestinal microbiota increases the risk of colon cancer, accelerates the aging of the intestinal mucosa, and inflammatory bowel disease (26, 31). Conventional clinical treatment of FC relies mainly on laxatives and prokinetic agents, but long-term use can induce drug dependence. Studies have confirmed that oral administration of specific probiotics can effectively improve constipation with little adverse effects and is gradually gaining favor among researchers (32–34). However, there are still significant challenges in the clinical application of probiotics for constipation treatment—specifically, the mechanisms of action of probiotics have not been fully clarified, the combined use of multiple strains is often required, and at the same time, strains effective for constipation are limited. Nevertheless, the development of new strains is of great help for the treatment of constipation. In this study, we explored a new probiotic strain, *L. acidophilus* C4, which can be used for the prevention of constipation.

We first confirmed the safety of the application of *L. acidophilus* C4. On the experimental dosage, *L. acidophilus* C4 intervention had no significant effect on the physiology of healthy mice as indicated by food intake, water intake, and body weight growth rate. Furthermore, it also had no impact on intestinal kinetics such as fecal water content and intestinal transit rate. FC is characterized by key symptoms such as reduced defecation frequency, hardened stools, and prolonged defecation time (23, 25, 35). Loperamide is a commonly used drug to induce FC. In this study, loperamide administration caused elongation of the time to first discharge of black stool, the decrease of the number of fecal pellets excreted within 6 h, fecal water content, and fecal wet weight which demonstrated the success in FC induction. Pretreatment by *L. acidophilus* C4 or *L. rhamnosus* LGG significantly reversed these pathological symptoms. Intestinal motility is mainly responsible for the movement of intestinal contents, and its peristalsis often slows down in patients with FC. While the intestinal motility was severely damaged in FC mice, *L. acidophilus* C4 pretreatment restored the intestinal motility as indicated by ink propulsion rate. In contrast, *L. acidophilus* C4 pretreatment had no impact on these in healthy mice, which suggests a specific role of *L. acidophilus* C4 in FC. Surprisingly, *L. rhamnosus* LGG did not have such a function in restoring the intestinal motility. Together, these results clearly demonstrated that *L. acidophilus* C4 pretreatment can alleviate loperamide-induced constipation in mice by reducing constipation and accelerating intestinal motility, which suggests *L. acidophilus* C4 could be a potent probiotic strain to prevent FC.

Intestinal barrier impairment contributed greatly to FC (23). ZO-1 and occludin are the two most important proteins constituting the intestinal tight junction proteins, and their expression levels can indirectly reflect the degree of integrity of the mucosal barrier. *L. acidophilus* C4 can upregulate the expression of ZO-1 and occludin in FC mice, which suggests a protective effect of C4 on intestinal barrier during FC. This is in line with a protective role of *L. acidophilus* C4 in intestinal motility as intestinal barrier is closely associated with intestinal motility (16). Dysregulated intestinal water content is a major reason for dry stools during FC (36, 37). AQPs in the colon play a crucial role in regulating fecal water content, and AQPs themselves are key proteins for modulating intestinal water homeostasis (37). Previous studies have found that AQP3 is mainly expressed in the mucosal epithelial cells of the colon (38). Meanwhile, research has shown that reduced function or expression of AQP3 leads to diarrhea due to decreased water absorption from the luminal side to the vascular side of the colon (36). We found that the *AQP3* mRNA was overexpressed in FC mice, while pretreatment of C4 downregulated

*AQP3* to a normal level. This may partly explain why *L. acidophilus* C4 increases the fecal water content.

Intestinal mucosal inflammation often accompanies constipation and may further impair intestinal motility. Herein, we found that *L. acidophilus* C4 significantly downregulates the expression of the pro-inflammatory factor IL-1β and upregulates the expression of the anti-inflammatory factor IL-10. Notably, the experimental findings regarding *L. rhamnosus* LGG presented in Fig. 6B are inconsistent with the established properties of this strain. As a well-characterized probiotic strain, *L. rhamnosus* LGG has been extensively documented to exert robust anti-inflammatory effects under a broad spectrum of physiological and pathological conditions (20, 39, 40). However, probiotic effects are highly strain-specific and context-dependent. The inflammatory milieu in chronic, low-grade constipation may differ from acute or systemic inflammatory models, potentially influencing probiotic efficacy. Furthermore, differences in bacterial metabolites, host interaction, and experimental protocols can lead to variable outcomes.

We conducted additional experiments to quantify lactic acid bacteria (LAB) colonization in mouse feces: At the experimental endpoint, fresh feces were collected, weighed, homogenized in sterile PBS, serially diluted, and plated on MRS agar under anaerobic conditions. Colony counts revealed a significant decrease in fecal LAB levels in the LOP group compared to the Ctrl group. Both *L. acidophilus* C4 and *L. rhamnosus* LGG interventions significantly restored these levels. This data, now included as Fig. S1.

We performed additional RT-qPCR analyses on colonic tissue. We focused on genes linking inflammation to metabolism, given their relevance in gut disorders. We found that loperamide-induced constipation significantly downregulated Srebf1 (a master regulator of lipid metabolism) and upregulated TNF-α (39–41). Intervention with *L. acidophilus* C4 or *L. rhamnosus* LGG effectively normalized the expression of both genes. This new data, presented in Fig. S2 and discussed in the text, provides deeper insight into how *L. acidophilus* C4 may ameliorate constipation-associated metabolic and inflammatory dysregulation.

To summarize, our study confirms that *L. acidophilus* C4 exerts preventive effects on FC and enhances intestinal motility, mainly by safeguarding the intestinal barrier and sustaining intestinal water balance.

## ACKNOWLEDGMENTS

This research was supported by Chongqing Natural Science Foundation (no. cstc2021jcyj-msxmX0158), National Natural Science Foundation of China (nos. 82100217 and 82302525) and Scientific and Technological Research Program of Chongqing Municipal Education Commission (no. KJQN202113201).

## AUTHOR AFFILIATIONS

[1]The Department of Infectious Diseases, the First Affiliated Hospital of Chongqing Medical University, School of Basic Medical Sciences, Chongqing Medical University, Chongqing, China
[2]State Key Laboratory of Infectious Disease Prevention and Control, National Institution for Communicable Disease Control and Prevention, Chinese Center for Disease Prevention and Control, Beijing, China

## AUTHOR ORCIDs

Yuan Wu http://orcid.org/0000-0002-9436-5929
He Lu http://orcid.org/0000-0001-6227-4996
Tianle Gu http://orcid.org/0000-0001-6163-9375
Zeng Tu http://orcid.org/0000-0002-0329-4097

## AUTHOR CONTRIBUTIONS

Jing Liu, Formal analysis, Investigation, Methodology.

## ETHICS APPROVAL

All protocols were approved by the Animal Ethics Committee of Chongqing Medical University (approval no.: IACUC-CQMU-2025-0371).

## ADDITIONAL FILES

The following material is available online.

### Supplemental Material

**Figures S1 and S2 (Spectrum02950-25-s0001.docx).** Figure S1: Fecal Lactobacillus colony counts in various treatment groups in mice. Figure S2: *L. acidophilus* C4 alleviates constipation-associated disorders of lipid metabolism levels.

### Open Peer Review

**PEER REVIEW HISTORY (review-history.pdf).** An accounting of the reviewer comments and feedback.

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
