## [Reviewer comments · Microbiology Spectrum]

Microbiology Spectrum

***Lactobacillus acidophilus* C4 ameliorates constipation in mice**

Zeng Tu, Han Yan, Qianying Jia, Jing Liu, Xinping Hua, Yihong Cui, Wenwen Jian, tanyu tan, Ochieng Oduor, Privillia Thomas, Mwansa Kawimbe, Yuan Wu, he lu, and Tianle Gu

Corresponding Author(s): Zeng Tu, Chongqing Medical University

Review Timeline:

Submission Date:	September 17, 2025
Editorial Decision:	November 6, 2025
Revision Received:	January 5, 2026
Editorial Decision:	January 25, 2026
Revision Received:	February 4, 2026
Accepted:	February 12, 2026

Editor: Samara Mattiello

Reviewer(s): The reviewers have opted to remain anonymous.

Transaction Report:

DOI: <https://doi.org/10.1128/spectrum.02950-25>

Re: Spectrum02950-25 (*Lactobacillus acidophilus* C4ameliorates constipation in mice)

Dear Dr. Zeng Tu:

Thank you for the privilege of reviewing your work. Below you will find my comments, instructions from the Spectrum editorial office, and the reviewer comments.

Revision Guidelines

Sincerely,
Samara Mattiello
Editor
Microbiology Spectrum

Reviewer #1 (Comments for the Author):

Functional constipation (FC) is a prevalent gastrointestinal disorder affecting individuals of all ages, particularly children and the elderly. Although chemical laxatives are generally effective, their use is often constrained by discomfort and the risk of drug dependence, highlighting the need for alternative interventions. Probiotics have been widely recognized for their role in promoting gastrointestinal (GI) health by modulating the gut microbiota. Extending this approach to the prevention and treatment of GI disorders, including FC, remains an active area of research emphasizing strain selection, formulation, and regimen

optimization.

In this preliminary study, the authors investigate the effects of a newly isolated *Lactobacillus acidophilus* C4 strain on FC prevention. In a loperamide-induced mouse model, oral administration of high-dose C4 (200 μ L of 1×10^9 CFU/mL, once or twice daily for 28 days) significantly alleviated constipation symptoms, enhanced intestinal motility, preserved mucosal barrier integrity, and reduced intestinal inflammation. Notably, the efficacy of C4 surpassed that of the reference *Lactobacillus rhamnosus* GG (LGG) strain, a well-characterized "gold standard" probiotic for gastrointestinal health. These findings suggest that *L. acidophilus* C4 may offer a more potent and streamlined probiotic regimen with potential advantages for manufacturing and regulatory development in FC management.

Suggestions to help the authors in both experimental design and data interpretation are listed below.

1. Lack of direct molecular evidence verifying the presence of the administered *Lactobacillus acidophilus* C4 strain in the gastrointestinal tract following treatment.
2. The persistence or transience of *Lactobacillus acidophilus* C4 and its influence on the overall gut microbiota composition are not discussed. These factors are critical for understanding the mechanistic basis and therapeutic translation of the probiotic's effects.
3. Figure 1: the inclusion of mouse images appears non-essential. If they do not contribute additional scientific value, consider removing them to maintain focus and clarity.
4. The statement on line 354 that C4 "exhibits a good safety" is overstated, as Figure 2 presents data from only a single-dose cohort. The same concern applies to lines 211-213. The safety conclusion should be restricted to the concentration and conditions tested.
5. Line 221 " Figs 2D and 2F should include 2E.
6. The LGG data in Figure 6B is surprising as literature in recent years supports LGG's anti-inflammatory effects. Please discuss these discrepant findings and provide potential explanations. Without such discussion or validation, it raises concern about assay reproducibility and interpretation.

Reviewer #2 (Comments for the Author):

In the "*Lactobacillus acidophilus* C4 ameliorates constipation in mice" by Yan et al. The authors explore the potential of *Lactobacillus acidophilus* C4 as a probiotic candidate to alleviate functional constipation and reduce intestinal inflammation using a mouse model. The study presents interesting findings but requires several clarifications and methodological improvements to strengthen the scientific validity of the work.

Introduction

Comment 1: Replace the term "flora" with "microbiota" throughout the manuscript to reflect current terminology.

Comment 2: Correct the sentence "...*Bifidobacterium* spp. and *Lactobacillus* spp.), represented by *Lactobacillus* spp...." since these are different genera and cannot be represented by one another.

Methodology

Comment 3: The text in lines 78-79 of the Introduction suggests that *L. acidophilus* C4 was newly isolated. However, Section 2.1 gives the impression that this strain was previously obtained. Please clarify whether this strain was newly isolated in the present study or sourced from a previous collection.

Comment 4: In line 93, both *L. acidophilus* C4 and *L. rhamnosus* GG are mentioned. The role of *L. rhamnosus* GG is unclear. Provide details about its origin and explain why it was included as part of this study.

Comment 5: Lines 114-115: The description of the control groups ("normal control [NC] and C4 control [C4N] after a 7-day acclimatization period") is confusing. Please clarify the grouping and treatment scheme.

Comment 6: The graphical overview of the experimental design does not match the written methodology and may cause confusion. Revise the figure to accurately reflect the experimental sequence.

Comment 7: The methodology does not describe the euthanasia process or how tissue samples were collected. This information should be included for reproducibility.

Comment 8: Add the IACUC approval number or equivalent ethical approval for the animal experiments.

Comment 9: The manuscript lacks evidence demonstrating that *L. acidophilus* C4 is a safe probiotic candidate. It is important to include genomic analysis to confirm the absence of virulence or antimicrobial resistance genes and to perform standard phenotypic safety tests such as hemolysis, bile salt tolerance, Caco-2 cell adhesion/invasion, cytotoxicity, and intestinal histology. Without these data, the claim that this strain is a probiotic candidate remains incomplete.

Comment 10: The selected genes analyzed for expression are insufficient to support probiotic activity. Additional genes related to immune modulation, intestinal barrier function, and SCFA metabolism should be evaluated.

Results and Discussion

Comment 11: The discussion omits important interpretation regarding Figure 6B. The result is unexpected since recent studies consistently show the anti-inflammatory effects of *L. rhamnosus* GG. The authors should address and discuss this discrepancy.

Comment 12: Replace all occurrences of "flora" with "microbiota."

Comment 13: The manuscript requires thorough English language revision by a fluent or professional editor to correct grammar and improve flow.

Reviewer Comments for the Author:

Functional constipation (FC) is a prevalent gastrointestinal disorder affecting individuals of all ages, particularly children and the elderly. Although chemical laxatives are generally effective, their use is often constrained by discomfort and the risk of drug dependence, highlighting the need for alternative interventions. Probiotics have been widely recognized for their role in promoting gastrointestinal (GI) health by modulating the gut microbiota. Extending this approach to the prevention and treatment of GI disorders, including FC, remains an active area of research emphasizing strain selection, formulation, and regimen optimization.

In this preliminary study, the authors investigate the effects of a newly isolated *Lactobacillus acidophilus* C4 strain on FC prevention. In a loperamide-induced mouse model, oral administration of high-dose C4 (200 μ L of 1×10^9 CFU/mL, once or twice daily for 28 days) significantly alleviated constipation symptoms, enhanced intestinal motility, preserved mucosal barrier integrity, and reduced intestinal inflammation. Notably, the efficacy of C4 surpassed that of the reference *Lactobacillus rhamnosus* GG (LGG) strain, a well-characterized “gold standard” probiotic for gastrointestinal health. These findings suggest that *L. acidophilus* C4 may offer a more potent and streamlined probiotic regimen with potential advantages for manufacturing and regulatory development in FC management.

Suggestions to help the authors in both experimental design and data interpretation are listed below.

1. Lack of direct molecular evidence verifying the presence of the administered *Lactobacillus acidophilus* C4 strain in the gastrointestinal tract following treatment.
2. The persistence or transience of *Lactobacillus acidophilus* C4 and its influence on the overall gut microbiota composition are not discussed. These factors are critical for understanding the mechanistic basis and therapeutic translation of the probiotic's effects.
3. Figure 1: the inclusion of mouse images appears non-essential. If they do not contribute additional scientific value, consider removing them to maintain focus and clarity.
4. The statement on line 354 that C4 “exhibits a good safety” is overstated, as Figure 2 presents data from only a single-dose cohort. The same concern applies to lines 211–213. The safety conclusion should be restricted to the concentration and conditions tested.
5. Line 221” Figs 2D and 2F should include 2E.
6. The LGG data in Figure 6B is surprising as literature in recent years supports LGG's anti-inflammatory effects. Please discuss these discrepant findings and

provide potential explanations. Without such discussion or validation, it raises concern about assay reproducibility and interpretation.

January,6 2026

Dear Editor,

Thank you very much for your kind help in handling our manuscript to Microbiology Spectrum 02950-25 (*Lactobacillus acidophilus* C4 ameliorates constipation in mice). We thank the reviewer for their positive evaluation of our work and for the constructive suggestions, which have been invaluable in strengthening our manuscript. We have addressed all points raised as detailed below. We marked all new changes with yellow highlighting in the text for your convenience. Please refer to the below point-to-point reply to each issue and the revised text for more details.

Reviewer #1 (Comments for the Author):

Functional constipation (FC) is a prevalent gastrointestinal disorder affecting individuals of all ages, particularly children and the elderly. Although chemical laxatives are generally effective, their use is often constrained by discomfort and the risk of drug dependence, highlighting the need for alternative interventions. Probiotics have been widely recognized for their role in promoting gastrointestinal (GI) health by modulating the gut microbiota. Extending this approach to the prevention and treatment of GI disorders, including FC, remains an active area of research emphasizing strain selection, formulation, and regimen optimization.

In this preliminary study, the authors investigate the effects of a newly isolated *Lactobacillus acidophilus* C4 strain on FC prevention. In a loperamide-induced mouse model, oral administration of high-dose C4 (200 μ L of 1×10^9 CFU/mL, once or twice daily for 28 days) significantly alleviated constipation symptoms, enhanced intestinal motility, preserved mucosal barrier integrity, and reduced intestinal inflammation. Notably, the efficacy of C4 surpassed that of the reference *L. rhamnosus* GG (LGG) strain, a well-characterized "gold standard" probiotic for gastrointestinal health. These findings suggest that *L. acidophilus* C4 may offer a more potent and streamlined probiotic regimen with potential advantages for manufacturing and regulatory development in FC management.

Suggestions to help the authors in both experimental design and data interpretation are listed below.

Comment 1: Lack of direct molecular evidence verifying the presence of the administered *Lactobacillus acidophilus* C4 strain in the gastrointestinal tract following treatment.

Response 1: We are grateful for the valuable insight raised by the reviewer. To address this concern, we attempted to specifically detect *L. acidophilus* C4 in the fecal samples of treated mice. However, despite conducting genome-wide analysis, we were unable to design strain-specific primer pairs for *L. acidophilus* C4. Therefore, we alternatively enumerated lactic acid bacteria (LAB) using de Man, Rogosa, and Sharpe (MRS) agar plates in an anaerobic incubator, as MRS medium is highly selective for LAB.

Colony counting data demonstrated that compared with the Ctrl group, the intestinal colonization level of LAB in the feces of the LOP group was significantly reduced. In contrast, intervention with either *L. acidophilus* C4 or *L. rhamnosus* GG notably restored the intestinal LAB colonization level in model mice. Although this result does not directly confirm the presence of the administered *L. acidophilus* C4 strain in the gastrointestinal tract post-treatment, it partially reflects the successful

intestinal colonization of the administered strain. We have added this result to the supplementary files to solidify our results. This data, now included as Supplementary Fig. S1.

Supplementary Fig. S1 Fecal *Lactobacillus* colony counts in various treatment groups in mice. At the experimental endpoint, freshly excreted feces from mice were serially diluted and spread onto bacterial culture plates, which were then anaerobically incubated for 24 hours prior to colony counting. Subsequent experiments included: (A) Colony counting and photography of bacterial culture plates; (B) Statistical analysis. Ctrl: Normal control group. LOP: Loperamide-induced model mice. LOP+C4: Loperamide-induced model mice prevented with *L. acidophilus* C4. LOP+LGG: Loperamide-induced model mice prevented with *L. rhamnosus* GG. Data are expressed as mean \pm standard deviation (SD) (n=9). Compared with the loperamide-induced model mouse group (****) $p < 0.0001$, compared with the normal control group, (####) $p < 0.0001$, is significant.

Comment 2: The persistence or transience of *Lactobacillus acidophilus* C4 and its influence on the overall gut microbiota composition are not discussed. These factors are critical for understanding the mechanistic basis and therapeutic translation of the probiotic's effects.

Response 2: We thank the reviewer for highlighting this critical aspect for mechanistic understanding. We acknowledge that characterizing the strain's persistence and its broader impact on the gut microbiota ecosystem is essential for therapeutic translation. We are currently conducting relevant experiments, including 16S rRNA gene sequencing, to investigate these dynamics. We will carry out a more comprehensive study planned for future work.

Comment 3: Figure 1: the inclusion of mouse images appears non-essential. If they do not contribute additional scientific value, consider removing them to maintain focus and clarity.

Response 3: As suggested, the non-essential mouse images have been removed from Figure 1 to enhance focus and clarity.

Comment 4: The statement on line 354 that C4 "exhibits a good safety" is overstated, as Figure 2 presents data from only a single-dose cohort. The same concern applies to lines 211-213. The safety conclusion should be restricted to the concentration and conditions tested.

Response 4: We agree with the reviewers that the safety conclusion was initially overstated. We have revised the text to precisely reflect the experimental conditions. The statements now read: Line 224-226 "These results indicated that *L. acidophilus* C4, at the dosage used in this experiment, had no significant impact on the physiological status indicators (body weight, food intake, and water

intake) of the mice"; line 365-367: "We first confirmed the good safety of *L. acidophilus* C4. On the experimental dosage, *L. acidophilus* C4 intervention had no significant effect on physiology of healthy mice as indicated by food intake, water intake, and body weight growth rate". All modified text has been highlighted in yellow in the revised manuscript.

Comment 5: Line 221" Figs 2D and 2F should include 2E.

Response 5: Thank you for catching this oversight. The text has been corrected to Line 234: "food or water intake (Figs 2D-2F)".

Comment 6: The LGG data in Figure 6B is surprising as literature in recent years supports LGG's anti-inflammatory effects. Please discuss these discrepant findings and provide potential explanations. Without such discussion or validation, it raises concern about assay reproducibility and interpretation.

Response 6: We thank the reviewers for their valuable comments regarding the apparent discrepancy of *L. rhamnosus* GG's effect on inflammatory factors in our model. We have expanded the discussion to address this. It is well-established that *L. rhamnosus* GG possesses anti-inflammatory properties in various contexts (1-3). However, probiotic effects are highly strain-specific and context-dependent. The inflammatory milieu in chronic, low-grade constipation may differ from acute or systemic inflammatory models, potentially influencing probiotic efficacy. Furthermore, differences in bacterial metabolites, host interaction, and experimental protocols can lead to variable outcomes. We have incorporated this into the manuscript, acknowledging the complexity of probiotic-host interactions and aligning our findings with the known variability in probiotic effects across different disease models.

References:

- 1) Ma W, Lian L, Guo L, Wu Y, Huang L. 2025. *Lactobacillus rhamnosus* Glory LG12 preventives loperamide-induced constipation in mice by modulating intestinal flora and metabolic pathways. *Front Microbiol* 16:1577799.
- 2) Wang T, Yan H, Lu Y, Li X, Wang X, Shan Y, Yi Y, Liu B, Zhou Y, Lü X. 2020. Anti-obesity effect of *Lactobacillus rhamnosus* LS-8 and *Lactobacillus crustorum* MN047 on high-fat and high-fructose diet mice base on inflammatory response alleviation and gut microbiota regulation. *Eur J Nutr* 59:2709–2728.
- 3) Wang G, Yang S, Sun S, Si Q, Wang L, Zhang Q, Wu G, Zhao J, Zhang H, Chen W. 2020. *Lactobacillus rhamnosus* Strains Relieve Loperamide-Induced Constipation via Different Pathways Independent of Short-Chain Fatty Acids. *Front Cell Infect Microbiol* 10:423.

Reviewer #2 (Comments for the Author):

In the "*Lactobacillus acidophilus* C4 ameliorates constipation in mice" by Yan et al. The authors explore the potential of *Lactobacillus acidophilus* C4 as a probiotic candidate to alleviate functional constipation and reduce intestinal inflammation using a mouse model. The study presents interesting findings but requires several clarifications and methodological improvements to strengthen the scientific validity of the work.

Comment 1: Replace the term "flora" with "microbiota" throughout the manuscript to reflect current terminology.

Response 1: Done.

Comment 2: Correct the sentence "...Bifidobacterium spp. and Lactobacillus spp.), represented by Lactobacillus spp...." since these are different genera and cannot be represented by one another.
Methodology

Response 2: Done.

Comment 3: The text in lines 78-79 of the Introduction suggests that *L. acidophilus* C4 was newly isolated. However, Section 2.1 gives the impression that this strain was previously obtained. Please clarify whether this strain was newly isolated in the present study or sourced from a previous collection.

Response 3: Thank you for your requesting clarification. This *L. acidophilus* C4 strain was previously isolated by us.. Its protective effects against dextran sulfate sodium (DSS)-induced colitis in mice have been previously published (1). In the current study, we investigate its new application in a constipation model. This clarification has been added to Section 2.1.

Reference

1) Liu Q, Jian W, Wang L, Yang S, Niu Y, Xie S, Hayer K, Chen K, Zhang Y, Guo Y, Tu Z. 2023. Alleviation of DSS-induced colitis in mice by a new-isolated *Lactobacillus acidophilus* C4. *Front Microbiol* 14:1137701.

Comment 4: In line 93, both *L. acidophilus* C4 and *L. rhamnosus* GG are mentioned. The role of *L. rhamnosus* GG is unclear. Provide details about its origin and explain why it was included as part of this study.

Response 4: Thank you for your valuable comment. We added this information in the introduction section (lines 127-134). "*L. rhamnosus* GG (standard strain, ATCC 53103) was used as a well-established positive control in this study (1-2). The constipation-alleviating efficacy of *L. rhamnosus* GG has been verified by numerous clinical studies, and its mechanisms of action (including regulating intestinal flora, promoting intestinal motility, etc.) are well-defined with high safety, making it a classic reference strain in the field of probiotic intervention for constipation. Therefore, including *L. rhamnosus* GG allowed for a direct comparative assessment of the potency of novel *L. acidophilus* C4."

References:

1) Gu Y, Qin X, Zhou G, Wang C, Mu C, Liu X, Zhong W, Xu X, Wang B, Jiang K, Liu J, Cao H. 2022. *Lactobacillus rhamnosus* GG supernatant promotes intestinal mucin production through regulating 5-HT4R and gut microbiota. *Food Funct* 13:12144–12155.

2) Gui L, Duan X, Wang H, Xie H, Zhang R, Jiang W, Tang W. 2025. *Lactobacillus rhamnosus* GG maintains gut microbiota stability and promotes intestinal adaptation via activated intestinal farnesoid X receptor signaling in short bowel syndrome. *Commun Biol* 8:816.

Comment 5: Lines 114-115: The description of the control groups ("normal control [NC] and C4 control [C4N] after a 7-day acclimatization period") is confusing. Please clarify the grouping and treatment scheme.

Response 5: We apologize for the confusion. The group nomenclature and experimental design have been clarified and standardized throughout the text and figures. The NC group was treated only with normal saline, while the C4N group was treated only with *L. acidophilus* C4; neither group received loperamide treatment. To comply with academic norms and avoid group confusion, we have adopted new nomenclature for all experimental mouse groups. Specifically, the NC group has been renamed as the Ctrl group, the C4N group as the C4 group, the MC group as the LOP group, the original C4 group as the LOP+C4 group, and the original LGG group as the LOP+LGG group.

Comment 6: The graphical overview of the experimental design does not match the written methodology and may cause confusion. Revise the figure to accurately reflect the experimental sequence.

Response 6: Thank you for your valuable comment. We fully agree with your suggestion regarding terminology standardization. We have revised the information presented in the figures and tables. Additionally, to ensure clarity and consistency in experimental grouping, we have adopted new nomenclature for all experimental mouse groups. Specifically, the NC group has been renamed as the Ctrl group, the C4N group as the C4 group, the MC group as the LOP group, the original C4 group as the LOP+C4 group, and the original LGG group as the LOP+LGG group.

Comment 7: The methodology does not describe the euthanasia process or how tissue samples were collected. This information should be included for reproducibility.

Response 7: Thank you for your valuable comment. We have supplemented the Materials and Methods section, which can be found in line 139-147 of the revised manuscript and highlighted in yellow. "*At the end of the experiment, mice were euthanized. Anesthetic compound Tribromoethanol (250 mg/kg) was administered intraperitoneally to the mice. After complete anesthesia, the mouse's head was firmly pressed downward with the thumb and index finger of the left hand; simultaneously, the tail was grasped and pulled backward forcefully, resulting in immediate death (verified by absence of heartbeat and dilated pupils). Gastrointestinal tissues and intestinal contents were collected for further analysis. All samples were stored at -80 °C for subsequent experiments. Animal carcasses were placed in yellow biohazard bags and stored in a dedicated 4 °C freezer for experimental animal remains.*"

Comment 8: Add the IACUC approval number or equivalent ethical approval for the animal experiments.

Response 8: In the Materials and Methods section (Section 2.3: Mouse model and experimental design), the approval number of the Institutional Animal Care and Use Committee (IACUC) has been included. This information can be found in line 115 of the revised manuscript and highlighted in yellow. " (*Approval No.: IACUC-CQMU-2025-0371*) ".

Comment 9: The manuscript lacks evidence demonstrating that *L. acidophilus* C4 is a safe probiotic candidate. It is important to include genomic analysis to confirm the absence of virulence or antimicrobial resistance genes and to perform standard phenotypic safety tests such as hemolysis, bile salt tolerance, Caco-2 cell adhesion/invasion, cytotoxicity, and intestinal histology. Without these data, the claim that this strain is a probiotic candidate remains incomplete.

Response 9: We thank the reviewer for this critical point. Data from our previous study demonstrates that *L. acidophilus* C4 exhibits robust *in vitro* probiotic properties (acid and bile tolerance, adhesion potential) (1). In the current *in vivo* study, we confirmed the absence of adverse effects on general health (body weight, intake). We acknowledge that a full safety profile is required for clinical translation for future work. We have marked the sentence segment "...which exhibits potent *in vitro* probiotic properties, including acid and bile tolerance and intestinal adhesion capability" in yellow, corresponding to lines 78–79 of the manuscript.

Supplementary Table S1 Determination of the intestinal colonization ability of *L. acidophilus* C4

Acid resistance and bile salt resistance of C4					
Strain		Acid-resistant survival rate		Bile-tolerant salt survival rate	
C4		53.57%		80%	

Surface hydrophobicity rate and surface self-aggregation rate of L. Acidophilus C4					
Experiment	Time (h)	OD _{A0}	OD _{A1}	Surface hydrophobicity	Surface self-cohesion
Surface hydrophobicity	0.25h	0.573	0.045	92.08%	—
	0h	1.476	1.476	—	0
	1h	1.476	1.449	—	1.81%
Surface self-cohesion	3h	1.476	1.409	—	4.52%
	5h	1.476	0.971	—	34.21%
	6h	1.476	1.03	—	30.22%
	24h	1.476	0.223	—	84.94%

Reference:

1) Liu Q, Jian W, Wang L, Yang S, Niu Y, Xie S, Hayer K, Chen K, Zhang Y, Guo Y, Tu Z. 2023.

Alleviation of DSS-induced colitis in mice by a new-isolated *Lactobacillus acidophilus* C4. *Front Microbiol* 14:1137701.

Comment 10: The selected genes analyzed for expression are insufficient to support probiotic activity. Additional genes related to immune modulation, intestinal barrier function, and SCFA metabolism should be evaluated.

Response 10: We agree that analyzing a broader panel of genes would strengthen the mechanistic insight. We performed additional RT-qPCR analyses on colonic tissue. We focused on genes linking inflammation to metabolism, given their relevance in gut disorders. We found that loperamide-induced constipation significantly downregulated *Srebf1* (a master regulator of lipid metabolism) and upregulated *TNF- α* . Intervention with *L. acidophilus* C4 or *L. rhamnosus* LGG effectively normalized the expression of both genes. This new data, presented in Supplementary Fig. S2 and discussed in the text, provides deeper insight into how *L. acidophilus* C4 may ameliorate constipation-associated metabolic and inflammatory dysregulation (1-3).

Supplementary Fig. S2 *L. acidophilus* C4 alleviates constipation-associated disorders of lipid metabolism levels. Mice were treated as Fig 1B, on day 27 of the experiment, mice were sacrificed and dissected, and colon tissues were harvested. The (A) *Srebf1* and (B) *TNF- α* mRNA levels were measured by RT-qPCR. Data are expressed as mean \pm standard deviation (SD) (n=6). Compared with the loperamide-induced model mouse group, (***) $p < 0.001$, (****) $p < 0.0001$, compared with the normal control group, (#) $p < 0.05$, (####) $p < 0.0001$, is significant.

References:

- 1) Eberlé D, Hegarty B, Bossard P, Ferré P, Foufelle F. 2004. SREBP transcription factors: master regulators of lipid homeostasis. *Biochimie* 86:839–848.
- 2) Ma W, Lian L, Guo L, Wu Y, Huang L. 2025. *Lactobacillus rhamnosus* Glory LG12 prevents loperamide-induced constipation in mice by modulating intestinal flora and metabolic pathways. *Front Microbiol* 16:1577799.
- 3) Wang T, Yan H, Lu Y, Li X, Wang X, Shan Y, Yi Y, Liu B, Zhou Y, Lü X. 2020. Anti-obesity effect of *Lactobacillus rhamnosus* LS-8 and *Lactobacillus crustorum* MN047 on high-fat and high-fructose diet mice based on inflammatory response alleviation and gut microbiota regulation. *Eur J Nutr* 59:2709–2728.

Results and Discussion

Comment 11: The discussion omits important interpretation regarding Figure 6B. The result is

unexpected since recent studies consistently show the anti-inflammatory effects of *L. rhamnosus* LGG. The authors should address and discuss this discrepancy.

Response 11:

We thank the reviewers for their valuable comments regarding the apparent discrepancy of *Lactobacillus rhamnosus* LGG's effect on inflammatory factors in our model. We have expanded the discussion to address this. It is well-established that *Lactobacillus rhamnosus* LGG possesses anti-inflammatory properties in various contexts (1-3). However, probiotic effects are highly strain-specific and context-dependent. The inflammatory milieu in chronic, low-grade constipation may differ from acute or systemic inflammatory models, potentially influencing probiotic efficacy. Furthermore, differences in bacterial metabolites, host interaction, and experimental protocols can lead to variable outcomes. We have incorporated this into the manuscript, acknowledging the complexity of probiotic-host interactions and aligning our findings with the known variability in probiotic effects across different disease models.

References:

- 1) Ma W, Lian L, Guo L, Wu Y, Huang L. 2025. *Lactobacillus rhamnosus* Glory LG12 preventives loperamide-induced constipation in mice by modulating intestinal flora and metabolic pathways. *Front Microbiol* 16:1577799.
- 2) Wang T, Yan H, Lu Y, Li X, Wang X, Shan Y, Yi Y, Liu B, Zhou Y, Lü X. 2020. Anti-obesity effect of *Lactobacillus rhamnosus* LS-8 and *Lactobacillus crustorum* MN047 on high-fat and high-fructose diet mice base on inflammatory response alleviation and gut microbiota regulation. *Eur J Nutr* 59:2709–2728.
- 3) Wang G, Yang S, Sun S, Si Q, Wang L, Zhang Q, Wu G, Zhao J, Zhang H, Chen W. 2020. *Lactobacillus rhamnosus* Strains Relieve Loperamide-Induced Constipation via Different Pathways Independent of Short-Chain Fatty Acids. *Front Cell Infect Microbiol* 10:423.

Comment 12: Replace all occurrences of "flora" with "microbiota."

Response 12: Done.

Comment 13: The manuscript requires thorough English language revision by a fluent or professional editor to correct grammar and improve flow

Response 13: The manuscript has undergone thorough professional language editing to correct grammatical errors and improve overall fluency and academic tone.

Re: Spectrum02950-25R1 (*Lactobacillus acidophilus* C4ameliorates constipation in mice)

Dear Dr. Zeng Tu:

Thank you for the privilege of reviewing your work. Below you will find my comments, instructions from the Spectrum editorial office, and the reviewer comments.

Dear Authors,

Thank you for addressing the reviewers' comments. Below you will find the remaining reviewer suggestions for your consideration.

Revision Guidelines

Sincerely,
Samara Mattiello
Editor
Microbiology Spectrum

Reviewer #1 (Comments for the Author):

Thank you for addressing most of our questions. Only a few typographical issues require attention:
1. Please replace "flora" with "microbiota" throughout the manuscript, e.g., line 70.

2. Correct the font size of "De Man, Rogosa and Sharpe" in line 97.
3. Capitalize "We" in line 422.

Reviewer #2 (Comments for the Author):

Thank you very much for taking the time to address the questions and suggestions provided to improve the clarity of the manuscript.

The authors have addressed most of the comments satisfactorily; however, a few additional points still require attention:

1. The formatting is inconsistent throughout the manuscript. Please ensure that margins, text alignment (justified vs. aligned), font type, and font size are consistent across all sections.
2. Please replace the term "flora" with microbiota throughout the manuscript to reflect current and appropriate terminology.

February,4 2026

Dear Editor,

Thank you very much for your kind help in handling our manuscript to Microbiology Spectrum 02950-25R2 (*Lactobacillus acidophilus* C4 ameliorates constipation in mice).

We have provided detailed explanations for the issues raised. For your convenience, we have highlighted all the new changes in the text in yellow. Please refer to the responses to the questions and the revised text below for more details.

Problem 1: 1. We noticed that there are significant overlaps in sections 2.6 and 2.7 under the Materials and Methods section. Please check and include appropriate citations.

Response 1: Thank you for pointing this out. We have made corrections to the semantic repetition issues in Sections 2.5, 2.6 and 2.7 of the Materials and Methods section. At the same time, we have added citations. The corrected parts have been highlighted in yellow.

Re: Spectrum02950-25R2 (*Lactobacillus acidophilus* C4ameliorates constipation in mice)

Dear Dr. Zeng Tu:

Your manuscript has been accepted, and I am forwarding it to the ASM production staff for publication. Your paper will first be checked to make sure all elements meet the technical requirements. ASM staff will contact you if anything needs to be revised before copyediting and production can begin. Otherwise, you will be notified when your proofs are ready to be viewed.

Sincerely,
Samara Mattiello
Editor
Microbiology Spectrum